# Correlation between Key Steps and Hydricity in CO$_2$ Hydrogenation Catalysed by Non-Noble Metal PNP-Pincer Complexes

Snehasis Moni and Bhaskar Mondal *

School of Chemical Sciences, Indian Institute of Technology Mandi, Mandi 175005, Himachal Pradesh, India
* Correspondence: bhaskarmondal@iitmandi.ac.in

**Abstract:** Transition metal-catalysed homogeneous hydrogenation of CO$_2$ to formate or formic acid has emerged as an appealing strategy for the reduction of CO$_2$ into value-added chemicals. Since the state-of-the-art catalysts in this realm are primarily based on expensive precious metals and require demanding reaction conditions, the design and development of economically viable non-noble metal catalysts are in great demand. Herein, we exploit the thermodynamic correlation between the crucial reaction steps of CO$_2$ hydrogenation, that is, base-promoted H$_2$-splitting and hydride transfer to CO$_2$ as a guide to estimate the catalytic efficiency of non-noble metal complexes possessing a ligand backbone containing a secondary amine as an "internal base". A set of three non-noble metal complexes, one bearing tri-coordinated PNP-pincer (**1$_{Mn}$**) and the other two based on tetra-coordinated PNPN-pincer (**2$_{Mn}$** and **3$_{Fe}$**), have been investigated in this study. The computational mechanistic investigation establishes the role of the "internal" amine base in heterolytically splitting the metal-bound H$_2$, a critical step for CO$_2$ hydrogenation. Furthermore, the thermodynamic correlation between the hydricity ($\Delta G^{\circ}_{H^-}$) of the in situ generated metal-hydride species and the free energy barrier of the two crucial steps could provide an optimal hydricity value for efficient catalytic activity. Based on the computational estimation of the optimal hydricity value, the tri-coordinated PNP-pincer complex **1$_{Mn}$** appears to be the most efficient among the three, with the other two tetra-coordinated PNPN-pincer complexes, **2$_{Mn}$** and **3$_{Fe}$**, showing promising hydricity values. Overall, this study demonstrates how the crucial thermodynamic and kinetic parameters for pincer-based complexes possessing an "internal base" can be correlated for the prediction of novel non-noble metal-based catalysts for CO$_2$ hydrogenation.

**Keywords:** CO$_2$ hydrogenation; non-noble metal; PNP pincer ligand; internal base; hydricity; DFT calculations

## 1. Introduction

The anthropogenic release of CO$_2$ due to the extensive combustion of fossil fuels creates environmental turmoil [1,2]. Over the past few decades, the drastic increase in atmospheric CO$_2$ concentration has triggered climate change, global warming, rising sea levels, ocean acidification, etc. [3–6] This motivates researchers to search for alternative fuels and renewable energy sources. In this direction, the utilization of CO$_2$ as a C1 source for value-added chemicals and fuels has attracted significant research attention in recent times owing to the low-cost, high abundance, relatively low toxicity, and recyclability of CO$_2$ [7,8]. Over the past twenty years, economically viable strategies have been developed through which atmospheric CO$_2$ can be converted into environmentally congenial fuel components [9,10]. However, the reduction of CO$_2$ poses tremendous challenges due to its thermodynamic stability and kinetic inertness [11]. The kinetic inertness of CO$_2$ demands efficient catalytic processes, which usually involve multiple-electron transfer coupled with protons or Lewis acids (e.g., transition metals) [9–11]. In this direction, a great amount of research has been devoted to developing both homogeneous and heterogeneous catalytic processes for converting CO$_2$ to useful chemicals and fuels [12,13].

Among the wide variety of homogeneous $CO_2$ reduction pathways [7,14–16], $CO_2$ hydrogenation becomes one of the most attractive ones as this pathway can offer a myriad range of value-added chemicals, such as formic acid, formate, formaldehyde, methanol, methane, etc. [17–19], and can serve as a crucial step for sustainable organic syntheses [20,21]. The elemental $CO_2$ hydrogenation process leading to formic acid, formate, or their derivatives is particularly one of the most efficient pathways of $CO_2$ conversion to useful chemicals [22–26]. Since the first report of homogeneous $CO_2$ hydrogenation by Inoue et al. [27], an appreciable number of noble and non-noble transition metal-based homogeneous $CO_2$ hydrogenation catalysts have been reported with noteworthy turnover numbers (TON) and turnover frequencies (TOF) [9,26]. Among the noble metal catalysts, an iridium-pincer catalyst $[IrH_3(PNP^iPr)]$ ($PNP^iPr$ = 2,6-$(CH_2P^iPr_2)_2C_5H_3N$) was reported to exhibit a remarkable highest-to-date TOF of 150,000 h$^{-1}$ for $CO_2$ hydrogenation to formate [28]. However, when it comes to the non-noble metal catalysts, the reactivity lacks far behind that of noble metals. The highest $CO_2$ hydrogenation reactivity with a TOF of 74,000 h$^{-1}$ has been reported with a cobalt(I)-phosphine catalyst, $[Co^I(dmpe)_2H]$ (dmpe = 1,2-bis(dimethylphosphino)ethane), in the presence of a very strong Verkade base [29]. Therefore, it is highly desirable to develop robust $CO_2$ hydrogenation catalysts based on environmentally benign and cost-effective non-noble metals that can operate in ambient conditions. The rational design and development of such catalysts necessitate an atomic-level, understating of the key reaction steps of $CO_2$ hydrogenation and crucial thermodynamic and kinetic factor(s) that dictate the overall reactivity.

Transition metal-catalyzed $CO_2$ hydrogenation usually follows a common reaction mechanism as shown in Scheme 1a [30]. The catalytic cycle consists of three distinct reactions steps, involving (i) base-promoted $H_2$-splitting to form a metal hydride species (**I1** → **I2**), (ii) nucleophilic attack of hydride (H$^-$) of the metal-hydride species (**I2**) undergoing hydride transfer from metal to $CO_2$ (**I3** → **I4**), and (iii) protonation of the terminal oxygen of formate (HCOO$^-$) and release of the formic acid (**I4** → **I5**). Previous reports showed that among these three steps, either $H_2$-splitting or hydride transfer can act as the rate-determining step (RDS) of the entire catalytic cycle depending on the nature of the catalyst and the corresponding metal-hydride complex [30–34]. This catalyst-dependent switching of the RDS was demonstrated experimentally using Ru(II) $[(\eta^6-C_6Me_6)-Ru^{II}(bpy)(H)]^+$ and Ir(III) $[(\eta^5-C_5Me_5)Ir^{III}(bpy)-(H)]^+$ (bpy = 2,2′-bipyridine) complexes, where the former exhibited an $H_2$-splitting RDS and the later showed a hydride transfer RDS [35].

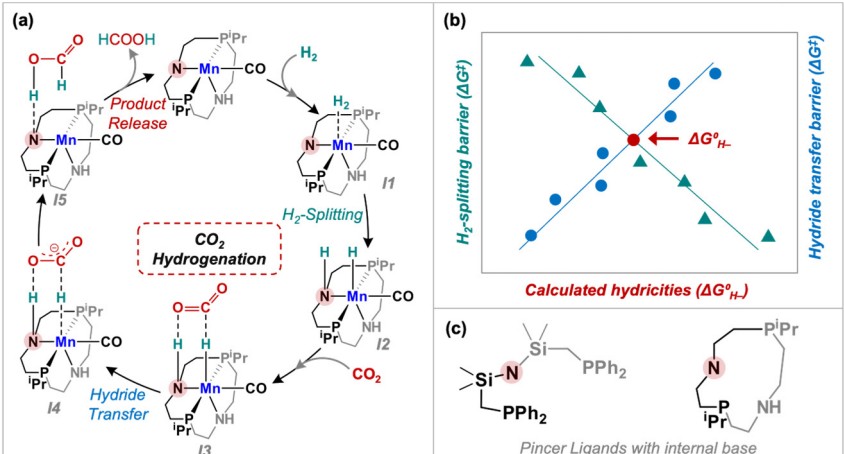

**Scheme 1.** (**a**) Established reaction mechanism and steps involved in homogeneous $CO_2$ hydrogenation. (**b**) Schematic representation of the correlation diagram between the two key reaction steps of $CO_2$ hydrogenation and hydricity. (**c**) Chemical structures of the tri- and tetra-coordinated pincer ligands.

It has been realized that the $H_2$-splitting step is driven by the strength of the M–H bond formation in the metal hydride; on the other hand, the hydride transfer is driven by the ability of the M–H bond to donate the $H^-$ to $CO_2$ [36,37]. As such, the electronic requirement of the two key steps is quite the opposite. Therefore, a delicate balance must be maintained between the two key steps of $CO_2$ hydrogenation, that is, $H_2$ splitting and hydride transfer for efficient execution of the reaction. As the two crucial steps are connected by the intermediacy of the M–H bond in the metal hydride intermediate (**I2**), the balance is controlled by the strength of the M–H bond (Scheme 1b). The M–H bond strength can be quantified using a thermodynamic parameter, called hydricity ($\Delta G_{H^-}^{\circ}$), which is a measure of the ability of the metal-hydride complex to donate its hydride. Understandably, a lower hydricity (less positive value) will ease the hydride donation, and the opposite is true for the $H_2$ splitting process. Using phosphine-based catalysts of the type $[M(H)(\eta^2\text{-}H_2)(PP_3{}^{Ph})]^{n+}$ (M = Fe(II), Ru(II), and Co(III); $PP_3{}^{Ph}$ = *tris*(2-(diphenylphosphino)phenyl)phosphine), Mondal et al. established that metal hydride species possessing relatively low hydricity exhibit base-promoted $H_2$-splitting RDS, whereas complexes with high hydricity undergo hydride transfer RDS [36]. In a subsequent report, the same group used hydricity as a guide to present rational design strategies for predicting efficient $CO_2$ hydrogenation catalysts based on non-noble metals [37]. These computational reports have been truly motivating in the rational design and development of non-noble metal-based catalysts for $CO_2$ hydrogenation to formate or formic acid.

One of the major challenges in base-assisted $CO_2$ hydrogenation reactions is the requirement of an "external" sacrificial base to promote the $H_2$-splitting step as well as the product release. This can be efficiently circumvented using pincer-based complexes featuring an "internal base" in-built into the pincer ligand. Leitner and coworkers reported how metal-ligand cooperation through a secondary amine (–N–) functionality in the Mn-PNP pincer complex (**1$_{Mn}$**) can promote the activation of pinacolborane during the hydroboration of carbon dioxide [38]. Later on, a computational mechanistic investigation by Lei and coworkers demonstrated the promising potential of complex **1$_{Mn}$** towards $CO_2$ hydrogenation to methanol [39]. Hazari et al. showcased how the Lewis acidic behavior of a PNP-pincer ligand's N–H moiety can be exploited for the Fe-catalyzed $CO_2$ hydrogenation to formate [40]. A similar Fe-pincer catalytic system was also used by Bernskoetter et al. for the $CO_2$ hydrogenation to methanol [41]. These reports present a highly promising prospect that the N-atom in the pincer chain (Scheme 1c) can be exploited as a base for the crucial $H_2$-splitting step and the corresponding form as a Lewis acid to stabilize the formate product (Scheme 1a). The tri-coordinated PNP-pincer ligands, although extremely efficient in inducing versatile catalytic reactivities in transition metal complexes, occasionally suffer from ligand dissociation in harsh reaction conditions. Very recently, Li et al. computationally predicted that a more rigid version of the pincer ligand, tetra-coordinated PNPN pincer (Scheme 1c), can be highly promising in the hydrogenolysis of polyurethanes [42]. The computationally predicted novel Mn- and Fe-based tetra-coordinated PNPN complexes (**2$_{Mn}$** and **3$_{Fe}$**) possessing an "internal base" appear highly promising toward $CO_2$ hydrogenation.

Herein, we investigate how the two crucial reaction steps of $CO_2$ hydrogenation to formate/formic acid are the base-promoted $H_2$-splitting and hydride transfer in the catalytic cycle of metal-pincer complexes featuring an "internal base" are correlated with the hydricity of the metal-hydride intermediate. Specifically, we have performed a mechanistic investigation and reaction energetics calculations on the tetra-coordinated PNPN-pincer complexes **2$_{Mn}$** and **3$_{Fe}$** along with the experimentally reported tri-coordinated Mn-PNP pincer complex **1$_{Mn}$** (Scheme 2) using density functional theory (DFT) calculations. Importantly, we have correlated the driving force ($\Delta G$) and reaction barrier ($\Delta G^{\ddagger}$) of the $H_2$-splitting and hydride transfer steps of all the complexes (**1**–**3**) with the calculated hydricities of their metal-hydride intermediates. Finally, based on the correlation plots, we predicted the catalytic potential of the non-noble metal PNP-pincer complexes for $CO_2$ hydrogenation.

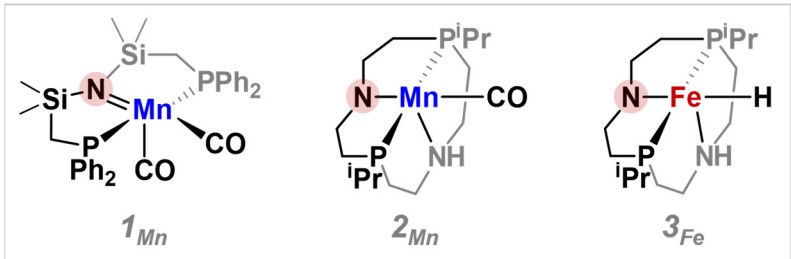

**Scheme 2.** PNP-pincer complexes (**1**–**3**) of the non-noble metals (Mn and Fe) possessing the N-atom as an "internal base" are investigated in this work.

## 2. Computational Details

All the geometry optimizations and harmonic vibrational frequency calculations have been performed at the density functional theory (DFT) level, employing the range-separated hybrid functional ωB97XD in conjunction with the LANL2DZ basis set and corresponding effective core potential (ECP) on the metals and Pople's double-ξ basis set 6-31G(d,p) on the main-group elements. The conductor-like continuum solvation model (CPCM) with tetrahydrofuran (THF, $\varepsilon$ = 7.4257) as solvent was employed during the geometry optimization. We also investigated whether there is any explicit solvent effect on the intrinsic reaction barrier and driving force using an explicit solvent (THF) molecule in the proximity of the reaction center. No noticeable change in the barrier and the driving force was observed due to the explicit solvent molecule (Figure S1 from Supplementary Materials). For a relatively accurate estimation of reaction energetics, a basis set combination of LANL2TZ and the corresponding ECP on the metals and the triple-ξ basis set 6-311++G(d,p) describing the main-group elements was used for the calculation of electronic energies at the ωB97XD level. The long-range corrected DFT functional ωB97XD has been reported in the literature to produce reliable results on the reaction energetics of metal-pincer complexes [32]. All the transition states were confirmed by connecting the reactants and products by intrinsic reaction coordinate (IRC) calculations and characterized by only one imaginary frequency. The enthalpies (H) and Gibbs free energies (G) were calculated at standard conditions, 298 K and 1 atm. All the calculations were performed using the Gaussian 16 suite of the quantum chemistry program [43]. A detailed description of the calculation of thermodynamic hydricities ($\Delta G^{\circ}_{H^-}$) [44–48] at the DFT level is presented in the Supporting Information.

## 3. Results and Discussion

The complete reaction free energy profile involving the reaction steps described in Scheme 1a for $CO_2$ hydrogenation catalyzed by the reported Mn(I)-PNP pincer complex **1**$_\text{Mn}$ leading to formic acid is presented in Figure 1. The energetics for the different reaction steps is calculated at the DFT-ωB97xD level. Prior to the $H_2$-splitting, the molecular hydrogen binds to the Mn-center in an endergonic manner (**I1**, $\Delta G$ = 10.6 kcal/mol) followed by the base-promoted splitting of the metal-bound $H_2$ to generate the metal-hydride intermediate (**I2**). The N-atom of the PNP-pincer ligand serves as the "internal base" to assist the $H_2$-splitting process. The overall process (**1**$_\text{Mn}$ → **I2**) appears to be thermoneutral ($\Delta G$ = 0.1 kcal/mol) and involves a moderate free-energy barrier of 14.6 kcal/mol through an $H_2$-splitting transition state (**TS1**). The net free energy barrier and free energy change for the $H_2$ splitting step are similar to that reported earlier by Lei et al. [39] Importantly, the intrinsic driving force, that is, from the metal-bound $H_2$ complex to the metal-hydride intermediate (**I1** → **I2**), is calculated to be 10.5 kcal/mol, and the corresponding free-energy barrier is only 4.0 kcal/mol (**I1** → **TS1**). The low intrinsic free energy barrier for the $H_2$-splitting step for **1**$_\text{Mn}$ is due to the large entropic penalty associated with the $H_2$ binding, as clearly seen from the 4.7 kcal/mol overall enthalpy barrier. A relatively large intrinsic driving force for the $H_2$-splitting process (**I1** → **I2**) for **1**$_\text{Mn}$ is indicative of a rather strong Mn–H bond in the metal-hydride intermediate **I2**.

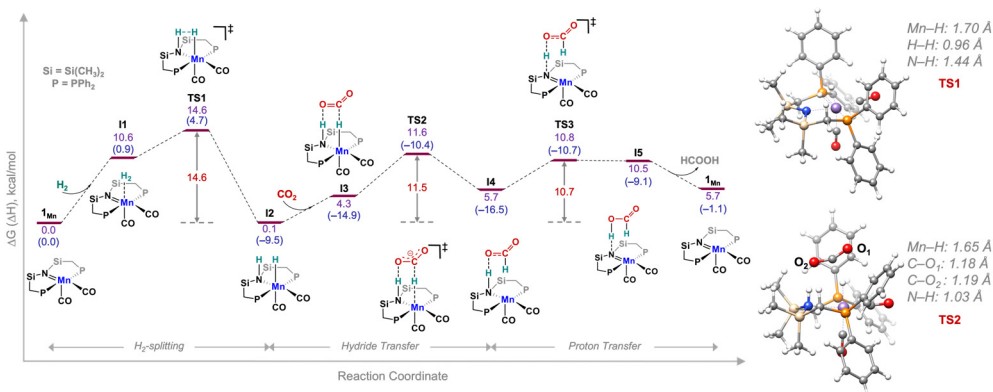

**Figure 1.** The free energy ($\Delta$G in kcal/mol) profile along with the transition states of the key steps of $CO_2$ hydrogenation to formic acid catalyzed by ($1_{Mn}$) at the $\omega$B97XD/[6-311++G(d,p)+LANL2TZ]/CPCM(THF) level of theory. Relative enthalpy ($\Delta$H in kcal/mol) values are presented in parentheses. ‡ symbol is used to denote transition states.

A moderately endergonic binding of $CO_2$ ($\Delta$G = 4.2 kcal/mol) to the metal hydride intermediate **I2** to form a $CO_2$ adduct **I3** precedes the hydride transfer step and the resulting intermediate **I3** undergoes hydride transfer to $CO_2$ to form a formate ($HCOO^-$)-bound intermediate **I4**. The overall free energy barrier for the hydride transfer step was calculated to be 11.5 kcal/mol (**I2** $\rightarrow$ **TS2**) with an intrinsic barrier of 7.3 kcal/mol (**I3** $\rightarrow$ **TS2**). The hydride transfer step for $1_{Mn}$ was calculated to be overall endergonic by 5.7 kcal/mol and intrinsically thermoneutral (**I3** $\rightarrow$ **I4**, $\Delta$G = 1.4 kcal/mol). Therefore, considering the intrinsic free energy barriers and driving forces, the $H_2$-splitting step appears to be energetically more feasible for $1_{Mn}$ as compared to the hydride transfer step.

The final step, involving a proton transfer from the "internal base", i.e., the N-atom of the PNP-pincer ligand to the formate ($HCOO^-$), leads to the formic acid-bound catalyst complex (**I5**) through an endergonic process (**I4** $\rightarrow$ **I5**, $\Delta$G = 4.8 kcal/mol). This proton transfer step involves an intrinsic barrier of 5.1 kcal/mol and an overall barrier of 10.7 kcal/mol. The final release of formic acid (HCOOH) and catalyst regeneration (**I5** $\rightarrow$ $1_{Mn}$) makes the overall process thermoneutral.

As evident from the overall free energy profile in Figure 1 for the $CO_2$ hydrogenation reaction catalyzed by $1_{Mn}$, base-promoted $H_2$-splitting appears to be the rate-determining step of the overall catalytic cycle. This result is consistent with the previous literature reports on the $CO_2$ hydrogenation reaction catalyzed by metal-pincer complexes (M = Ir, Fe, Co, Mn) [32,39]. As our main focus remains on the correlation between the free energy barrier of the key steps and the hydricity of the metal-hydride intermediate, a fair comparison between different catalysts needs reaction barriers to be ascertained on the basis of an identical reference point. To this end, calculation of the free energy barrier using the infinitely separated species (catalyst + $H_2$) typically predict a substantial entropy change of ~10 kcal/mol due to the loss of translational and rotational degrees of freedom, as was also observed for the three catalysts under investigation [36,37,49]. Therefore, we use the reaction barrier for the key steps, the intrinsic barrier, using the reactant complex (**I1**) as the reference point for all the catalysts $1_{Mn}$, $2_{Mn}$, and $3_{Fe}$. A similar strategy of using such "intrinsic barriers" to establish the correlation between the key steps and hydricity in $CO_2$ hydrogenation reactions has been reported in the literature [36,37].

After analyzing the key steps' energetics for the reported Mn-PNP complex, we investigated the same for the computationally predicted tetra-coordinated PNPN-pincer complexes of Mn ($2_{Mn}$) and Fe ($3_{Fe}$). We particularly focus on the reaction energetics of the two key steps only, i.e., base-promoted $H_2$-splitting, and hydride transfer in the following discussion. Figure 2 represents the reaction energetics of the key steps exhibited by $2_{Mn}$. The complex $2_{Mn}$ inherits the lowest energy conformer, where the CO ligand coordinates to one of the tetragonal positions (Figure S2). The catalytically active form of $2_{Mn}$ is calculated

to possess a triplet ground state ($^3$$\mathbf{2_{Mn}}$, Figure 2) that is 4.6 kcal/mol lower in free energy as compared to the singlet. However, the triplet state ($^3$$\mathbf{2_{Mn}}$) does not bind to the molecular hydrogen to undergo the crucial $H_2$-splitting process and must experience a triplet to singlet spin cross to perform the reaction. Such a spin cross can introduce some uncertainty in estimating the overall free energy barrier, which could be avoided through "intrinsic" free energy barriers. The singlet $\mathbf{2_{Mn}}$ experiences a thermoneutral $H_2$ binding to the Mn-center and, consequently, a moderately lower $H_2$-splitting free energy barrier of 8.4 kcal/mol (**I1** → **TS1**, Figure 2), leading to the metal-hydride intermediate **I2** through the $H_2$-splitting transition state **TS1**. This calculated free energy barrier is very similar to the earlier reported intrinsic $H_2$-splitting barrier [42]. The thermoneutral binding of the molecular $H_2$ to the Mn center of $\mathbf{2_{Mn}}$ may be attributed to the strong interaction between the Mn center and $H_2$, as evident from the appreciably high binding enthalpy of −6.8 kcal/mol. The overall $H_2$-splitting process has a thermodynamic driving force of 6.2 kcal/mol in terms of free energy. Unlike the $H_2$ binding, the binding of $CO_2$ to the Mn-hydride intermediate in the subsequent step is moderately endergonic by 5.7 kcal/mol, which is due to the relatively weak adduct between the metal-hydride intermediate **I2** and $CO_2$. The overall hydride transfer barrier is only 7.4 kcal/mol, including the $CO_2$ binding step, which implies a marginally low intrinsic hydride transfer barrier (1.7 kcal/mol) for the process. The net hydride transfer process is calculated to be highly exergonic by 8.5 kcal/mol (**I2** → **I4**) with an intrinsic driving force of 14.2 kcal/mol.

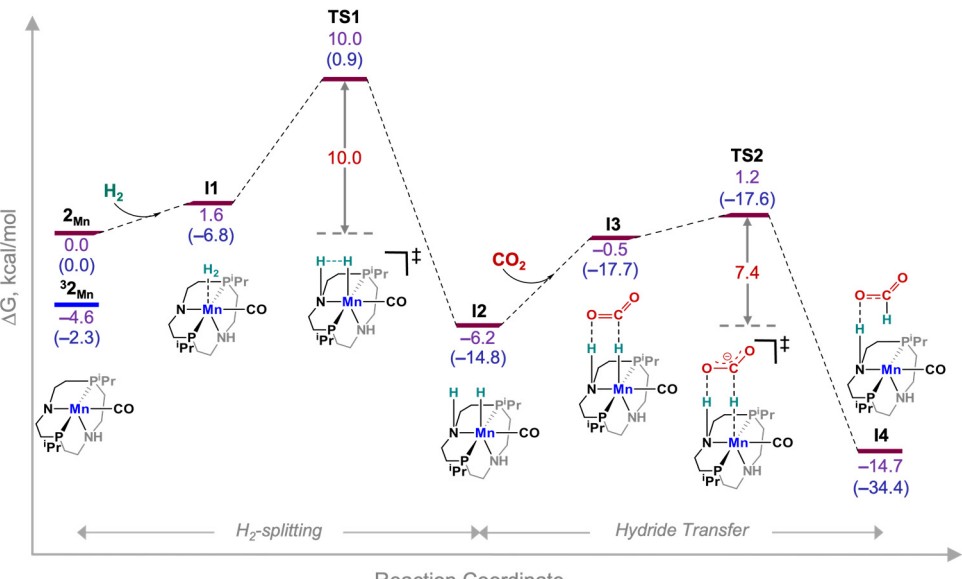

**Figure 2.** The free energy profile for the two key steps of $CO_2$ hydrogenation to formic acid catalyzed by complex $\mathbf{2_{Mn}}$ obtained at the $\omega$B97XD/[6-311++G(d,p)+LANL2TZ]/CPCM(THF) level of theory. ‡ symbol is used to denote transition states.

Similar to complex $\mathbf{2_{Mn}}$, $\mathbf{3_{Fe}}$ is a tetra-coordinated PNPN-pincer complex that was also theoretically predicted during the investigation of the hydrogenolysis of polyurethanes [42]. Complex $\mathbf{3_{Fe}}$ possesses the lowest energy conformer, where the H ligand coordinates to one of the tetragonal positions (Figure S2). Similar to complex $\mathbf{2_{Mn}}$, the coordinatively unsaturated catalytically active form of $\mathbf{3_{Fe}}$ possesses a triplet ground state ($^3$$\mathbf{3_{Fe}}$, Figure 3) that also does not bind to the molecular $H_2$. Therefore, the intrinsic reaction barriers for the key steps are obtained using singlet species. $\mathbf{3_{Fe}}$ undergoes $H_2$-splitting with an overall free energy barrier of 9.9 kcal/mol, which is similar to that of its Mn congener. The intrinsic barrier for this step for $\mathbf{3_{Fe}}$ appears to be just 2.5 kcal/mol lower, 7.4 kcal/mol (**I1** → **TS1**), as compared to the overall barrier. The subsequent hydride transfer step (**I2** → **TS2**) for $\mathbf{3_{Fe}}$ involves a lower barrier of 8.1 kcal/mol with a negligible intrinsic barrier of 2.0 kcal/mol

($I3 \rightarrow TS2$). Both the $H_2$-splitting and hydride transfer steps were calculated to have appreciable intrinsic driving forces of 10.4 and 13.9 kcal/mol, respectively.

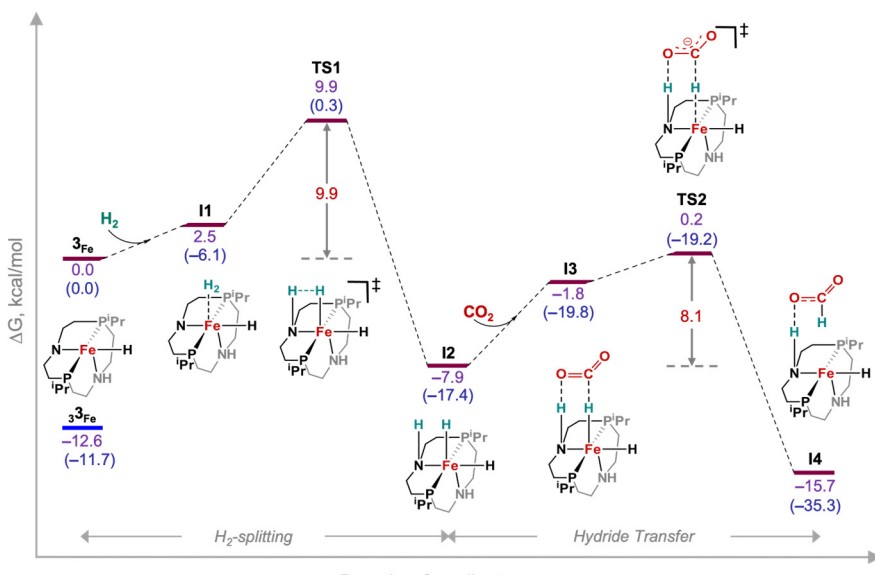

**Figure 3.** The free energy profile for the two key steps of $CO_2$ hydrogenation to formic acid catalyzed by complex $3_{Fe}$ obtained at the $\omega$B97XD/[6-311++G(d,p)+LANL2TZ]/CPCM(THF) level of theory. ‡ symbol is used to denote transition states.

As reported earlier, the two key steps of $CO_2$ hydrogenation, which are base-promoted $H_2$-splitting and hydride transfer to $CO_2$ have opposite electronic requirements [30,36,37]. The metal-hydride intermediate (**I2**) is the bridge and helps attain the balance between the two steps through the M–H bond. The strength of the M–H bond is quantified by the thermodynamic parameter, hydricity ($\Delta G^{\circ}_{H^-}$), measuring the metal-hydride intermediate to donate its hydride. Therefore, correlations between the hydricity of **I2** and the free energy barrier of $H_2$-splitting and hydride transfer would regulate the efficiency of a catalyst towards $CO_2$ hydrogenation. To estimate the catalytic potential of complexes $1_{Mn}$, $2_{Mn}$, and $3_{Fe}$, we correlated the calculated hydricities of their in situ generated metal-hydride intermediates (**I2**) with the corresponding barriers of $H_2$-splitting and hydride transfer. The computational protocol for calculating the hydricity is described in the Supporting Information. The computed hydricity of $HCOO^-$ matches exactly with the experimental value of 43.0 kcal/mol, which supports the computational method applied for calculating the hydricity in this work. Using the computational method of hydricity calculation as described in the Supporting Information, the calculated hydricities of $1_{Mn}$, $2_{Mn}$, and $3_{Fe}$ are 55.2, 47.1, and 48.7 kcal/mol, respectively.

Earlier reports by Mondal et al. [36,37] on phosphine-based $CO_2$ hydrogenation catalysts clearly demonstrated that the calculated $\Delta G^{\circ}_{H^-}$ of the metal-hydride species is linearly well-correlated with the barriers ($\Delta G^{\ddagger}$) of both $H_2$-splitting and hydride transfer. As one can expect, $\Delta G^{\circ}_{H^-}$ was found to be negatively and positively correlated with the barriers of $H_2$-splitting and hydride transfer, respectively. This corroborates well with the opposite electronic requirements of the two key steps of $CO_2$ hydrogenation, that is, a stronger hydricity will promote the $H_2$ splitting step and a weaker hydricity will promote the hydride transfer step. However, the opposite electronic requirements of the two key steps are not appropriately reflected when we examine the correlation plots between the "overall" free energy barrier and calculated hydricity. Specifically, the $H_2$ splitting step shows an unexpected positive correlation (Figure S4a). On the other hand, the notion of the opposite electronic requirement of the two key steps of $CO_2$ hydrogenation is nicely followed for the current complexes when we consider the "intrinsic" free energy barrier. For instance, the complex $1_{Mn}$ possessing the highest hydricity ($\Delta G^{\circ}_{H^-}$ = 55.2 kcal/mol) among three

exhibits a very low intrinsic barrier for the $H_2$-splitting (4.0 kcal/mol). Whereas, for the other two complexes, $2_{Mn}$ and $3_{Fe}$, possessing a relatively lower hydricity, the Intrinsic $H_2$-splitting barrier is much higher, at 8.4 and 7.4 kcal/mol, respectively (Table 1). The opposite was found true for the hydride transfer steps for the three complexes. Complex $1_{Mn}$, with the highest hydricity, traverses a much higher intrinsic hydride transfer barrier (7.3 kcal/mol) as compared to complexes $2_{Mn}$ and $3_{Fe}$ (1.7 and 2.0 kcal/mol, respectively) (Table 1). We observed a negative correlation between $\Delta G^{\circ}_{H^-}$ and $\Delta G^{\ddagger}$ of $H_2$-splitting ($R^2 = 1.00$, Figure S5a), and a positive correlation between $\Delta G^{\circ}_{H^-}$ and $\Delta G^{\ddagger}$ of hydride transfer ($R^2 = 0.98$, Figure S5b) for complexes $1_{Mn}$, $2_{Mn}$, and $3_{Fe}$. Thus, the use of an intrinsic free energy barrier, as also suggested in the earlier reports on $CO_2$ hydrogenation in ref. [30,36,37], gives thermodynamic correlations appropriately reflecting the electronic requirements of the key reaction steps. On merging the correlations between calculated hydricities of the metal-hydride intermediate (**I2**) and free energy barriers for the $H_2$-splitting and hydride transfer together, we could obtain an optimal value in the hydricity ($\Delta G^{\circ}_{H^-}$) scale, at 52.7 kcal/mol, that the catalysts under investigation need to strike for an optimal $CO_2$ hydrogenation reactivity (Figure 4).

**Table 1.** Calculated hydricity and intrinsic barriers for $H_2$ splitting and the hydride transfer step.

| Species | Calculated Hydricity ($\Delta G^{\circ}_{H^-}$, kcal/mol) | Intrinsic $H_2$-Splitting Barrier ($\Delta G^{\ddagger}$, kcal/mol) | Intrinsic Hydride Transfer Barrier ($\Delta G^{\ddagger}$, kcal/mol) |
|---|---|---|---|
| $1_{Mn}$ | 55.2 | 4.0 | 7.3 |
| $2_{Mn}$ | 47.1 | 8.4 | 1.7 |
| $3_{Fe}$ | 48.7 | 7.4 | 2.0 |

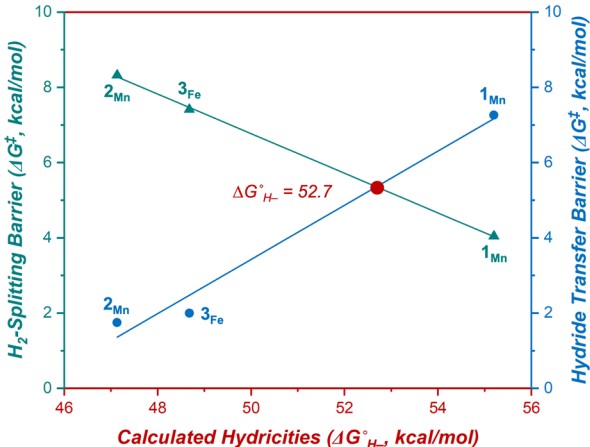

**Figure 4.** Correlation plot between the intrinsic free energy barrier of the key steps of $CO_2$ hydrogenation and calculated hydricities of the metal-hydride intermediates obtained at the DFT-ωB97xD level.

The combined correlation plot presented in Figure 4 delivers an estimation of the optimal value for hydricity for Mn- and Fe-based complexes possessing an "internal base" in the tri- or tetra-coordinated PNP-pincer ligand framework. The metal-hydride intermediate generated from the Mn-pincer complex $1_{Mn}$ reported by Leitner et al. appears to possess a hydricity value (55.2 kcal/mol) closest to the optimal value (52.7 kcal/mol); therefore, it can be considered as the most promising $CO_2$ hydrogenation catalyst among the three metal-pincer complexes. The Mn- and Fe-based tetra-coordinated PNPN-pincer complexes $2_{Mn}$ and $3_{Fe}$ possess hydricities of 47.1 and 48.1 kcal/mol, respectively, which are about 5–6 kcal/mol off from the estimated optimum value. Considering the uncertainty limit of the DFT-predicted hydricity and reaction barriers, the new tetra-coordinated PNPN-pincer complexes appear promising in homogeneous $CO_2$ hydrogenation.

## 4. Conclusions and Outlook

Non-noble metal-based pincer complexes possessing secondary amine as an "internal base" have been explored for homogeneous $CO_2$ hydrogenation in this study. Tricoordinated PNP-pincer complexes are well-known for $CO_2$ hydrogenation; however, their tetra-coordinated congeners need attention towards $CO_2$ hydrogenation. The current DFT-based mechanistic investigation and reaction energetics calculations on one tri-coordinated PNP-pincer Mn complex ($\mathbf{1_{Mn}}$) and two tetra-coordinated PNPN-pincer complexes ($\mathbf{2_{Mn}}$ and $\mathbf{3_{Fe}}$) reveal the crucial role of the secondary amine moiety in-built in the pincer backbone in the heterolytic $H_2$-splitting. The elegant correlation between the calculated hydricity of the metal-hydride intermediate and the free energy barrier of the crucial $H_2$-splitting and hydride transfer steps have been used to estimate the catalytic potential of the three pincer complexes. Specifically, the correlation plot provided an optimum value for hydricity, which is 52.7 kcal/mol. Indeed, the reported Mn-PNP complex $\mathbf{1_{Mn}}$ appears to be the most promising in terms of the balancing hydricity (55.2 kcal/mol) of the corresponding metal-hydride species. The other two tetra-coordinated PNPN-pincer complexes, $\mathbf{2_{Mn}}$ and $\mathbf{3_{Fe}}$, with relatively lower hydricity of their metal-hydride species, can be considered as potential catalysts for $CO_2$ hydrogenation within the uncertainty limit of the DFT calculations. This study provides a foundation for exploring the catalytic potential of a wide range of metal-pincer complexes towards $CO_2$ hydrogenation and will certainly guide novel catalyst design and development in this direction.

**Supplementary Materials:** The following supporting information can be downloaded at: https://www.mdpi.com/article/10.3390/catal13030592/s1, Figure S1: Role of explicit solvent; Figure S2: Conformational analysis; Tables S1 and S2 and Figure S3: Details of hydricity calculation; Table S3 and Figure S4: Individual correlation between hydricity and the overall reaction-free energy barrier; Figure S5: Individual correlation between hydricity and the intrinsic reaction-free energy barrier; Table S4: Optimized Cartesian coordinates for all the species. References [30,36,37,44] are cited also in Supplementary Materials.

**Author Contributions:** Conceptualization, B.M. and S.M.; methodology, S.M.; formal analysis, B.M. and S.M.; investigation, S.M.; data curation, S.M.; writing—original draft preparation, B.M. and S.M; supervision, B.M.; funding acquisition, B.M. All authors have read and agreed to the published version of the manuscript.

**Funding:** This research was funded by the Science and Engineering Research Board (SERB) start-up research grant (SRG/2020/000691) and IIT Mandi Seed Grant (IITM/SG/ABP/76).

**Data Availability Statement:** Not applicable.

**Acknowledgments:** S.M. is thankful to the Ministry of Education (MoE), Govt. of India for the research fellowship. The high-performance computing (HPC) facility at IIT Mandi is acknowledged for its computational resources. We thank the reviewers for their valuable comments and suggestions.

**Conflicts of Interest:** The authors declare no conflict of interest.

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
