# Peer review of "Correlation between Key Steps and Hydricity in CO2 Hydrogenation Catalysed by Non-Noble Metal PNP-Pincer Complexes"

_catalysts, doi:10.3390/catal13030592_

Round 1
Reviewer 1 Report
This is a DFT study of key steps in the CO2 hydrogenation catalysed by three earth abundant metal pincer complexes, calling special attention to the relation between activation barriers and hydricities of the metal-hydride intermediates. Overall the work is well motivated, presented and discussed (also in context with computational and experimental studies from the literature).
However, while the results are interesting in principle, I have two major reservations concerning the interpretation and overall conclusion: Firstly, in the correlation plots (e.g. Fig. 4) the H2 splitting barriers apparently are calculated relative to the preceding H2 complexes. However, these are all essentially unbound, and in my opinion these barriers should be calculated relative to the separated reactants (or actually the MARI, see second major comment below). Therefore in all cases H2 splitting would be the RDS and the advocated design principle of optimal hydricity rather moot.
Secondly, the computed reaction profiles stop at the point of the formate intermediates. They should be completed by including the proton transfer step and release of the product, formic acid, restoring the active catalysts (ideally with the barriers for proton transfer if these TSs can be located – it may well be that this step proceeds continuously uphill with no barrier on the PES). This is needed to assess the overall barrier of the whole cycle under turnover, in the spirit of Shaik and Kozuch's energy span model. If product release from the formate intermediates is endergonic, it would raise the total barrier (i.e. the energy span between most abundant reaction intermediate, MARI, and highest-energy TS, HETS) beyond the barriers indicated in the figures, further invalidating the concept of optimal hydricity.
In addition, I have a number of methodological comments:
(a) The spin states for the metal complexes should be specified. I assume the Mn carbonyl complexes are clear singlets, but for the complexes with the [FeH]+ fragment this is much less obvious. High -and intermediate-spin complexes need to be calculated (in particular for the coordinatively unsaturated complexes) in order to identify the electronic ground state.
(b) The LANL2DZ basis used on the metals is deplorably small. While this might be OK for the geometry optimisations this will lead to a large imbalance in the single points with the diffuse TZP-type basis on the ligands (and probably large BSSEs for H2 complex formation). The single points should be repeated with a bigger basis on the metals, e.g. the triple-zeta bases developed for the LANL potentials (J. Chem. Theory Comput. 2008, 4, 1029).
(c) The complexes derived from 2_Mn and 3_Fe are depicted with CO and H, respectively, in the tetragonal plane and an NH moiety from the pincer ligand at the apex. Has it been ensured that these are more stable than the corresponding isomers with apical CO or H ligands?
Because additional calculations are required and the whole purpose of the paper may need to be revised I think publication at this point would seem premature. If the methodological issues can be addressed, and a convincing repurposing of the results offered, I encourage submission of a suitably revised manuscript.
Author Response
A detailed response to reviews is attached.

Reviewer 2 Report
The manuscript reports a computational study of non-noble metal-based complexes" as homogeneous catalysts for CO2 hydrogenation with an "internal base." Using DFT results, the authors suggest a correlation between hydricity and free energy barriers in key steps. The article is well written and the conclusions are supported by the calculations. The only doubt about the proposed mechanism is the total absence of solvent and in particular the role of an explicit water molecule in this mechanism. A water molecule could reduce the bond strength in both TS1 and I2, and this specific interaction could increase the free energy barrier. The presence of water should be considered to present a CO2 reduction mechanism closer to the experimental conditions. Therefore, before proposing the publication of this manuscript, the authors should discuss this point.
Author Response
A detailed response to reviews is attached

Round 2
Reviewer 1 Report
I don't quite agree with the justification for using the (unbound) H2 complexes as reference, rather than the separated reactants. If anything, the different entropies for the different binding modes in the H2 complexes would argue against such a practice – the entropy changes on going from these H2 complexes to the H2-splitting-TSs may be smaller than those between separated reactants and TSs, but the latter should be more systematic (because they should not be affected by the binding modes in the intermediate adducts). But if it is common practice to do this (backed with a reference) I will not object. However, in order to avoid the impression that the authors had a preconceived idea about this correlation with hydricities and have picked the quantity that happens to give the best correlation, correlations with the true H2-splitting barrier should be reported and discussed in addition (i.e. that calculated relative to the separated reactants), so that the readers can make their own judgment. These correlations can be deposited in the ESI.
I must say that I find the complete neglect of the triplet ground states disturbing. It is true that this is a difficult issue for DFT and I agree that a deeper study of this is beyond the scope of the present paper. However, at the very least the spin states used must be stated and it must be conceded in which cases lower ones were found. These lower triplet states should be included in Figures 2 and 3. If all H2 adducts are singlets, the author's correlation using "intrinsic" H2-splitting barriers won't be affected (though the "true" ones discussed above may). If large S-T gaps are predicted that would seem to invalidate the findings (through very large H2-splitting barriers) a brief discussion of that issue is in order after all. One could argue that the high fraction of HF exchange in the range-separated hybrid may artificially stabilise the higher spin (e.g. from work by M. Reiher on Fe spin-crossover complexes it is known that functionals with lower such fractions tend to perform better for the spin-state energies, viz. his B3LYP* functional). By the way, that uncertainty about the true H2-splitting barrier due to the spin state of the active catalyst could be taken as further argument for using the intrinsic ones in the correlations.
As far as I can see, my other comments have been addressed satisfactorily. With the minor revisions as detailed above, I am happy to recommend acceptance.
Author Response
The response to the review is attached

Reviewer 2 Report
The authors modified the manuscript with the inclusion of all suggestions of both reviewers. The quality of the study is definitively improved, and they dissolved my doubts about the role of an explicit solvent molecule on the CO2 reduction mechanism. Thus, I suggest the publication in the present form.
Author Response
The response to the review is attached
